# LEARNING TO COMMUNICATE USING CONTRASTIVE LEARNING

## ABSTRACT

Communication is a powerful tool for coordination in multi-agent RL. Inducing an effective, common language has been a difficult challenge, particularly in the decentralized setting. In this work, we introduce an alternative perspective where communicative messages sent between agents are considered as different incomplete views of the environment state. Based on this perspective, we propose to learn to communicate using contrastive learning by maximizing the mutual information between messages of a given trajectory. In communication-essential environments, our method outperforms previous work in both performance and learning speed. Using qualitative metrics and representation probing, we show that our method induces more symmetric communication and captures task-relevant information from the environment. Finally, we demonstrate promising results on zero-shot communication, a first for MARL. Overall, we show the power of contrastive learning, and self-supervised learning in general, as a method for learning to communicate.

## 1 INTRODUCTION

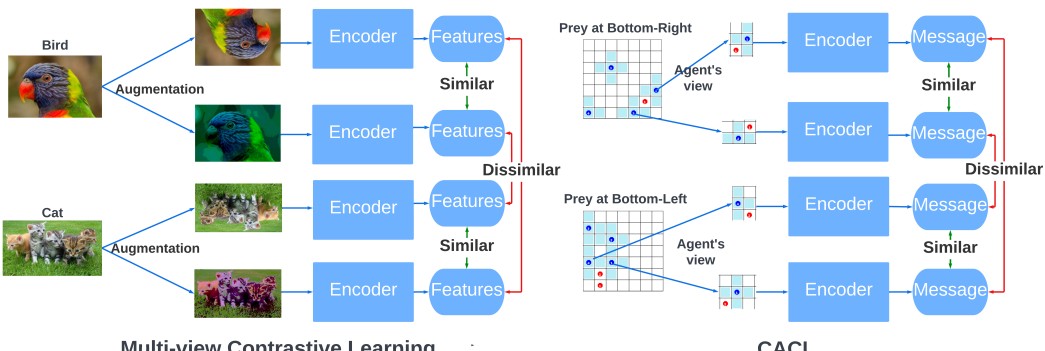

Figure 1: Multi-view contrastive learning and CACL, contrastive learning for multi-agent communication. In multi-view learning, augmentations of the original image or "views" are positive samples to contrastively learn features. In CACL, different agents' views of the same environment states are considered positive samples and messages are contrastively learned as encodings of the state.

Communication between agents is a key capability necessary for effective coordination among agents in partially observable environments. In multi-agent reinforcement (MARL) (Sutton & Barto, 2018), agents can use their actions to transmit information (Grupen et al., 2020) but continuous or discrete messages on a communication channel (Foerster et al., 2016), also known as linguistic communication (Lazaridou & Baroni, 2020), is more flexible and powerful. To successfully communicate, a speaker and a listener must share a common language with a shared understanding of the symbols being used (Skyrms, 2010; Dafoe et al., 2020). Learning a common protocol, or emergent communication (Wagner et al., 2003; Lazaridou & Baroni, 2020), is a thriving research direction but many works focus on simple, single-turn, sender-receiver games (Lazaridou et al., 2018; Chaabouni et al., 2019). In more visually and structurally complex MARL environments (Samvelyan et al.,

2019), existing approaches often rely on centralized learning mechanisms by sharing models (Lowe et al., 2017) or gradients (Sukhbaatar et al., 2016).

However, a centralized controller is impractical in many real-world environments (Mai et al., 2021; Jung et al., 2021) and centralized training with decentralized execution (CTDE) (Lowe et al., 2017) may not perform better than purely decentralized training (Lyu et al., 2021). Furthermore, the decentralized setting is more flexible and requires fewer assumptions about other agents, making it more realistic in many real-world scenarios (Li et al., 2020). The decentralized setting also scales better, as a centralized controller will suffer from the *curse of dimensionality*: as the number of agents it must control increases, there is an exponential increase in the amount of communication between agents to process (Jin et al., 2021). Hence, this work explores learning to communicate in order to coordinate agents in the decentralized setting. In MARL, this means each agent will have its own model to decide how to act and communicate, and no agents share parameters or gradients.

Normal RL approaches to decentralized communication are known to perform poorly even in simple tasks (Foerster et al., 2016). The main challenge lies in the large space of communication to explore, the high variance of RL, and a lack of common grounding to base communication on (Lin et al., 2021). Earlier work leveraged how communication influences other agents (Jaques et al., 2018; Eccles et al., 2019) to learn the protocol. Most recently, Lin et al. (2021) proposed agents that autoencode their observations and simply use the encodings as communication, using the shared environment as the common grounding. We propose to use the shared environment and the knowledge that all agents are communicating to ground a protocol. If, like Lin et al. (2021), we consider our agents' messages to be encodings of their observations then agents in similar states should produce similar messages. This perspective leads to a simple method based on contrastive learning to ground communication.

Inspired by the literature in representation learning that uses different "views" of a data sample (Bachman et al., 2019), for a given trajectory, we propose that an agent's observation is a "view" of some environment states. Therefore, different agents' messages are encodings of different "views" of the same underlying state. From this perspective, messages within a trajectory should be more similar to each other than to messages from another trajectory. We visually show our perspective in Figure 1. We propose that each agent use contrastive learning between sent and received messages to learn to communicate, which we term Communication Alignment Contrastive Learning (CACL).

We experimentally validate our method in three communication-essential environments and empirically show how our method leads to improved performance and speed, outperforming state-of-the-art decentralized MARL communication algorithms. To understand CACL's success, we propose a suite of qualitative and quantitative metrics. We demonstrate that CACL leads to more symmetric communication, allowing agents to be more mutually intelligible. By treating our messages as representations, we show that CACL's messages capture task-relevant semantic information about the environment better than baselines. Finally, we look at zero-shot cooperation with partners unseen at training time, a first for MARL communication. Despite the difficulty of the task, we demonstrate the first promising results in this direction. Overall, we argue that self-supervised learning is a powerful direction for multi-agent communication.

## 2 RELATED WORK

Learning to coordinate multiple RL agents is a challenging and unsolved task where naively applying single-agent RL algorithms often fails (Foerster et al., 2016). Recent approaches focus on agents parameterized by neural networks (Goodfellow et al., 2016) augmented with a message channel so that they can develop a common communication protocol (Lazaridou & Baroni, 2020). To solve issues of non-stationarity, some work focuses on centralized learning approaches that globally share models (Foerster et al., 2016), training procedures (Lowe et al., 2017), or gradients (Sukhbaatar et al., 2016) among agents. This simplifies optimization issues can still be sub-optimal (Foerster et al., 2016; Lin et al., 2021). This also violates independence assumptions, effectively modelling the multi-agent scenario as a single agent (Eccles et al., 2019).

This work focuses on independent, decentralized agents and non-differentiable communication. In previous work, Jaques et al. (2018) propose a loss to influence other agents but require explicit and complex models of other agents and their experiments focus on mixed cooperative-competitive

scenarios. Eccles et al. (2019) build on this and add biases to each agent's loss function that separately encourage positive listening and positive signaling. Their method is simpler but requires task-specific hyperparameter tuning to achieve reasonable performance and still underperforms in sensory-rich environments (Lin et al., 2021). Our work is closest to Lin et al. (2021), who leverage autoencoding as their method to learn a message protocol. Agent learn to reconstruct their observations and simply communicate their autoencoding. The authors find that it outperforms previous methods while being algorithmically and conceptually simpler. Our setup differs from Eccles et al. (2019); Lin et al. (2021) by using continuous instead of discrete messages. This choice is standard in contrastive learning (Chopra et al., 2005; He et al., 2020; Chen et al., 2020a) and common in embodied multi-agent communication (Sukhbaatar et al., 2016; Singh et al., 2018; Jiang & Lu, 2018; Das et al., 2019). As well, our representation learning task requires no extra learning parameters that are discarded at test time, whereas Lin et al. (2021) discard their decoder network.

Autoencoding is a form of generative self-supervised learning (SSL) (Doersch et al., 2015). We propose to use another form of SSL, contrastive learning (Chen et al., 2020a), as the basis for learning communication. We are motivated by recent work that achieves state-of-the-art representation learning on images using contrastive learning methods (Chen et al., 2020b) and leverages multiple "views" of the data. Whereas negative samples are simply different images, positive samples are image data augmentations or "views" of the original image (Bachman et al., 2019). Since our setup includes supervised labels, we base our method on SupCon (Supervised Contrastive Learning) (Khosla et al., 2020) which modifies the classic contrastive objective to use account for multiple positive samples. Also related is Dessì et al. (2021) who propose discrete two-agent communication as a contrastive learning task, we do the opposite and leverage contrastive learning for multi-agent communication.

## 3 PRELIMINARIES

We base our investigations on decentralized partially observable Markov decision processes (Dec-POMDPs) with *N* agents to describe a *fully cooperative multi-agent task* (Oliehoek & Amato, 2016). A Dec-POMDP consists of a tuple $G = \langle S, A, P, R, Z, \Omega, n, \gamma \rangle$. $s \in S$ is the true state of the environment. At each time step, each agent $i \in N$ chooses an action $a \in A^i$ to form a joint action $a \in A \equiv A^1 \times A^2 ... \times A^N$. It leads to an environment transition according to the transition function $P(s'|s, a^1, ...a^N) : S \times A \times S \rightarrow [0, 1]$. All agents share the same reward function $R(s, a) : S \times A \rightarrow \mathbb{R}$. $\gamma \in [0, 1)$ is a discount factor. As the environment is partially observable, each agent $i$ receives individual observations $z \in Z$ based on the observation function $\Omega^i(s) : S \rightarrow Z$.

We denote the environment trajectory and the action-observation history (AOH) of an agent $i$ as $\tau_t = s_0, a_0, ....s_t, a_t$ and $\tau_t^i = \Omega^i(s_0), a_0^i, ....\Omega^i(s_t), a_t^i \in T \equiv (Z \times A)^*$ respectively. A stochastic policy $\pi(a^i|\tau^i) : T \times A \rightarrow [0, 1]$ conditions on AOH. The joint policy $\pi$ has a corresponding action-value function $Q^\pi(s_t, a_t) = \mathbb{E}_{s_{t+1:\infty}, a_{t+1:\infty}}[R_t|s_t, a_t]$, where $R_t = \sum_{i=0}^{\infty} \gamma^i r_{t+i}$ is the discounted return. $r_{t+i}$ is the reward obtained at time $t + i$ from the reward function $R$.

To account for communication, similar to Lin et al. (2021), at each time step $t$, an agent $i$ takes an action $a_t^i$ and produces a message $m_t^i = \Psi^i(\Omega^i(s_t))$ after receiving its observation $\Omega^i(s_t)$ and messages sent at the previous time step $m_{t-1}^{-1}$, where $\Psi^i$ is agent $i$'s function to produce a message given its observation and $m_{t-1}^{-1}$ refers to messages sent by agents other than agent $i$. The messages are continuous vectors of dimensionality $D$.

## 4 METHODOLOGY

We propose a different perspective on the message space used for communication. At each time step $t$ for a given trajectory $\tau$, a message $m_t^i$ of an agent $i$ can be viewed as an incomplete view of the environment state $s_t$ because it is a function of the environment state as formulated in section 3. Naturally, messages of all the agents $a_t$ are different incomplete perspectives of $s_t$. To ground decentralized communication, we hypothesize that we could leverage this relationship between messages from similar states to encourage consistency and proximity of the messages across agents. Specifically, we propose maximizing the mutual information using contrastive learning which aligns the message space by pushing messages from similar states closer together and messages of different

states further apart. As a heuristic for state similarity, we consider a window of timesteps within a trajectory to be all similar states i.e. positive samples of each other. To guarantee dissimilar negative samples (Schroff et al., 2015), we use states from other trajectories as negatives.

We extend the recent supervised contrastive learning method SupCon (Khosla et al., 2020) to the MARL setting by considering multiple trajectories during learning. We refer to this loss formulation as *Communication Alignment Contrastive Learning (CACL)*. In this case, we consider messages within a trajectory to be different views of the same data sample with the same label.

Let $H$ be a batch of trajectories with messages $M$. Let $M_\tau$ be the messages in trajectory $\tau$. For an agent $i$, let $m_t^i \in M_\tau$ be its message at time $t$ and other messages in trajectory $\tau$ be $A_\tau(m_t^i) \equiv \{m' \in M_\tau : m' \neq m_t^i\}$. Therefore, positives for a message $m_t^i$ given a timestep window $w$ are $P(m_t^i) \equiv \{m_{t'}^j \in A_\tau(m_t^i) : t' \in [t - w, t + w]\}$. Formally, the contrastive loss is:

$$L_{CACL} = \sum_{m \in M} \frac{-1}{|P(m)|} \sum_{m_p \in P(m)} \log \frac{\exp(m \cdot m_p / \eta)}{\sum_{m_a \in M \setminus m} \exp(m \cdot m_a / \eta)} \tag{1}$$

Where $\eta \in \mathbb{R}^+$ is a scalar temperature and $|P(m)|$ is the cardinality.

Practically, each agent has a replay buffer that maintains a batch of trajectory data containing messages received during training to compute the *CACL* loss. Following Khosla et al. (2020), messages are normalized before the loss computation and a low temperature (i.e. $\eta = 0.1$) is used as it empirically benefits performance and training stability. The total loss for each agent is a reinforcement learning loss $L_{RL}$ using the reward to learn a policy (but not message head) and a separate contrastive loss $L_{CACL}$ to learn just the message head, formulated as follows:

$$L = L_{RL} + \kappa L_{CACL} \tag{2}$$

where $\kappa \in \mathbb{R}^+$ is a hyperparameter to scale the *CACL* loss.

## 5 EXPERIMENTS AND RESULTS

### 5.1 EXPERIMENTAL SETUP

We evaluate our method on three multi-agent environments with communication channels. Given the limited information each agent observes themselves, agents are encouraged to meaningfully communicate in order to improve task performance.

**Predator-Prey**: A variant of the classic game (Benda et al., 1986; Barrett et al., 2011) based on Koul (2019) where 4 agents (i.e. predators) have the cooperative goal to capture 2 randomly-moving prey by surrounding each prey with more than one predator. We devise a more difficult variation where agents are required to entirely surround a prey on all four sides for it to be captured and they cannot see each other in their fields of view. Therefore, it is essential for agents to communicate their positions and actions in order to coordinate their attacks. We evaluate each algorithm with episodic rewards during evaluation episodes.

**Find-Goal**: Proposed by Lin et al. (2021), agents' goal is to reach the green goal location as fast as possible in a grid environment with obstacles. We use 3 agents and ,at each time step, each agents observes a partial view of the environment centered at its current position. Unlike in Lin et al. (2021), we use a field of view of $3 \times 3$ instead of $5 \times 5$ to make the problem harder. Each agents receives an individual reward of 1 for reaching the goal and an additional reward of 5 when all of them reach the goal. Hence, it is beneficial for an agent to communicate the goal location once it observes the goal. As in Lin et al. (2021), we measure performance using episode length. An episode ends quicker if agents can communicate goal locations to each other more efficiently. Hence, a method has better performance if it has shorter episode lengths.

**Traffic-Junction**: Proposed by Sukhbaatar et al. (2016), it consists of A 4-way traffic junction with cars entering and leaving the grid. The goal is to avoid collision when crossing the junction. We use 5 agents with a vision of 1. Although not necessary, given the limited vision in agents,

communication could help in solving the task. We evaluate each algorithm with success rate during evaluation episodes.

All results are averaged over 12 evaluation episodes and over 6 random seeds. More details of the environments and parameters can be found in appendix A.1.

## 5.2 TRAINING DETAILS

We compare CACL to the state-of-the-art independent, decentralized method, autoencoded communication (AEComm; Lin et al., 2021), which grounds communication by reconstructing encoded observations. We also compare to baselines from previous work: independent actor critic without communication (IAC) and positive listening (PL; Eccles et al., 2019) which encourages agents to act differently when receiving different messages. We do not include the positive signalling loss (Eccles et al., 2019) because extending it to continuous messages is non-trivial but note that AEComm outperforms it in the discrete case (Lin et al., 2021). We also compare to DIAL (Foerster et al., 2016) which learns to communicate through differentiable communication and is therefore decentralized but not independent.

All methods use the same architecture based on the IAC algorithm with n-step returns and asynchronous environments (Mnih et al., 2016). Each agent has an encoder for observations and received messages. For methods with communication, each agent has a communication head to produce messages based on encoded observations. For policy learning, a GRU (Cho et al., 2014) is used to generate a hidden representation from a history of observations and messages. Agents use the hidden state for their the policy and value heads, which are 3-layer fully-connected neural networks. We perform spectral normalization (Gogianu et al., 2021) in the penultimate layer for each head to improve training stability . The architecture is shown in Figure 6 and hyperparameters are further described, both in Appendix A.2.

## 5.3 TASK PERFORMANCE

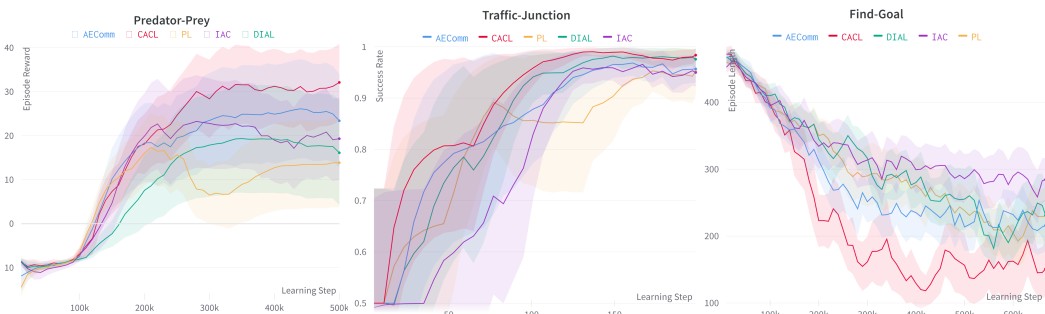

Figure 2: CACL (red) outperforms all other methods on Predator-Prey (left), Traffic-Junction (center) and Find-Goal (right). Predator-Prey shows evaluation reward, higher is better. Traffic-Junction plots the percent of successful episodes, higher is better. Find-Goal plots the length of the episode until the goal is reached, lower is better. Standard errors are plotted as shaded areas and the performance curves are smoothed by a factor of 0.5. .

We run all methods on Predator-Prey, Find-Goal, and Traffic-Junction and plot results in Figure 2. Our proposed method CACL outperforms all baseline methods in terms of both final performance and learning speed and, consistent with previous results (Lin et al., 2021), AEComm is the strongest baseline. The largest performance increase from CACL is in FindGoal where partial observability is most prominent because of agents' small field-of-view which makes communication more necessary (hence why IAC performs worst). These results show the effectiveness self-supervised methods for learning communication in the fully-decentralized setting, as they both outperform DIAL which, notably, backpropogates gradients through other agents. Furthermore, it demonstrates CACL's contrastive learning as a more powerful alternative to AEComm's autoencoding for coordinating agents with communication.

Table 1: Success rate in Predator-Prey: the percentage of final evaluation runs that captured no prey, one prey, or both prey. Average and standard deviation over 6 random seeds.

|  | No-Prey | One-Prey | Two-Preys |
|---|---|---|---|
| IAC | $36.67\% \pm 7.50$ | $20.00\% \pm 4.70$ | $48.33\% \pm 6.83$ |
| DIAL | $63.33\% \pm 7.56$ | $3.33\% \pm 1.24$ | $33.33\% \pm 7.86$ |
| PL | $50.00\% \pm 8.33$ | $0.00\% \pm 0.00$ | $51.67\% \pm 8.63$ |
| AEComm | $41.67\% \pm 7.48$ | $11.67\% \pm 2.79$ | $51.67\% \pm 7.60$ |
| CACL (Ours) | $\mathbf{33.33\% \pm 7.86}$ | $0.00\% \pm 0.00$ | $\mathbf{68.33\% \pm 8.07}$ |

To give practical context to our reward curves, we assess the algorithms from the perspective of task completion. In Predatory-Prey, we compute the percentages of evaluation episodes that capture no-prey, one-prey and two-preys, where capturing two preys is a successful. We average over 6 random seeds and, as shown in Table 1, CACL outperforms all baselines and not only solves the complete task more robustly, but also completely fails less frequently.

## 5.4 PROTOCOL SYMMETRY

To explain CACL's improved performance over the baselines, we hypothesize that it induces a more consistent, communal language that is shared among agents. More specifically, we consider a language's consistency to be how similarly agents communicate (i.e., sending similar messages) when faced with the same observations. A consistent protocol can reduce the optimization complexity since agents only need to learn one protocol for the whole group and it also makes agents more mutually intelligible.

To evaluate consistency, we measure protocol symmetry (Graesser et al., 2019) so if an agent swaps observations and trajectory with another agent, it should produce a similar message as what the other agent produced. We extend this metric from previous work to the continuous, embodied case and measure the pairwise cosine similarities of messages sent by different agents for the same observation. Let $\binom{N}{k}$ denote the set of all $k$-agent subset of a set of $N$ agents. Given a trajectory $\tau$ and $\{t \in T\}$ as a set of time steps of $\tau$, protocol symmetry ($protocol\_sym$) is written as:

$$protocol\_sym(\tau) = \frac{1}{|T|} \sum_{i \in T} \frac{1}{|N|} \sum_{i \in N} \frac{1}{|\binom{N}{k}|} \sum_{j,k \in \binom{N}{k}} \frac{m_j \cdot m_j}{\|m_j\|\|m_k\|} \tag{3}$$

Therefore, a more consistent protocol has higher symmetry. We swap agent trajectory and observations and compute this metric over 10 sampled evaluation episodes for 6 random seeds, and show results in Table 2.

Table 2: Protocol symmetry across environments, average and standard deviation over 10 episodes and 6 random seeds. CACL consistently learns the most symmetric protocol.

|  | DIAL | PL | AEComm | CACL (Ours) |
|---|---|---|---|---|
| Predator-Prey | $0.66 \pm 0.07$ | $0.66 \pm 0.06$ | $0.89 \pm 0.01$ | $\mathbf{0.95 \pm 0.01}$ |
| FindGoal | $0.50 \pm 0.05$ | $0.49 \pm 0.04$ | $0.85 \pm 0.02$ | $\mathbf{0.92 \pm 0.01}$ |
| Traffic Junction | $0.69 \pm 0.01$ | $0.61 \pm 0.04$ | $0.80 \pm 0.01$ | $\mathbf{0.98 \pm 0.002}$ |

The self-supervised methods (CACL and AEComm) clearly outperform the others (DIAL and PL) implying that SSL is better for learning consistent representations in decentralized MARL. Furthermore, CACL's protocol is very highly symmetric, clearly outperforming all others. Each AEComm agent autoencodes their own observation without considering the messages of other agents, leading to the formation of multiple protocols between agents. In contrast, CACL induces a common protocol by casting the problem in the multi-view perspective and implicitly aligning agents' messages. The possible correlation between protocol symmetry and overall performance and speed further indicates the benefits of learning a common language in the decentralized setting.

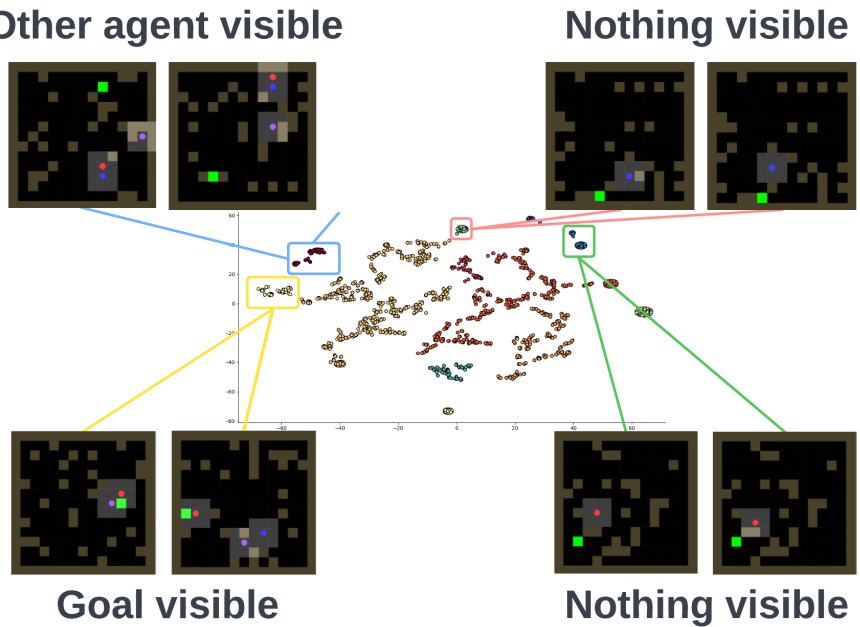

Figure 3: DBSCAN (Ester et al., 1996) clustering results of messages produced by CACL after reduced in dimensions using t-SNE (Van der Maaten & Hinton, 2008). Exemplary clusters are shown with their corresponding observational patterns. Specifically, two clusters correspond to messages sent when the goal is visible and another agent is visible respectively. The other two clusters of when only individual agents are visible.

## 5.5 PROTOCOL REPRESENTATION PROBING

To further investigate how informative our protocols are, we propose a suite of qualitative and quantitative representation probing tests based on message clustering and classification, respectively. We perform these tests on the protocols learned in the Find-Goal environment.

Similar to Lin et al. (2021), we perform message clustering on messages generated from 10 evaluation episodes to qualitatively assess whether CACL can learn an informative protocol. The messages are first compressed to a dimension of 2 using t-SNE (Van der Maaten & Hinton, 2008) and then clustered using DBSCAN (Ester et al., 1996). We look at each cluster's messages and their corresponding observations to extract any patterns and semantics captured. As shown in Figure 3, we observe a cluster of messages for observations when the goal is visible and a cluster for observations when another agent is visible. Two clusters correspond to agents seeing neither the goal nor another agent. This indicates that CACL learns to compress meaningful, task-relevant information in messages and allows agents to reasonably learn this semantic information.

Table 3: Classification results of the two probing tests in the Find-Goal environment, comparing all methods with communication. All methods perform similarly in the easier Goal Visibility Test while CACL outperforms the baselines significantly in the more difficult Goal Location Test.

|  | DIAL | PL | AEComm | CACL (Ours) |
|---|---|---|---|---|
| Goal Visibility | $99.45\% \pm 2.68$ | $98.87\% \pm 0.67$ | $99.75\% \pm 0.04$ | $97.75\% \pm 0.69$ |
| Goal Location | $68.15\% \pm 1.76$ | $78.31\% \pm 2.39$ | $76.14\% \pm 3.36$ | $91.28\% \pm 1.71$ |

To quantitatively evaluate the informativeness of learned protocols, we propose to treat messages as representations and follow literature in representation learning for RL (Lazaridou et al., 2018; Anand et al., 2019) to learn a classifier on top of the messages. Since FindGoal is focused on reaching a goal, intuitively, agents should communicate whether they have found the goal and, if so, where other agents should go to reach the goal. Therefore, we propose to probe the goal visibility and goal location. The former uses the messages to classify whether the goal is visible in observations or not

(i.e. a binary classification). The latter uses messages where the goal is visible in the observations to classify the general location of the goal (i.e. a 5-class classification: Top-Left, Top-Right, Bottom-Left, Bottom-Right and Middle). Goal location is more difficult to predict than goal visibility as it requires detailed, spatial information. We use 30 evaluation episodes per method to generate messages for our experiments but different methods may have different numbers of messages that are acceptable for our probing task (e.g. a limited number of messages where the goal is visible for predicting goal location). To ensure fair comparison, we choose an equal number of samples per class (i.e. ,positive/negative, 5-class location) for all methods and use a 70%/30% random split for training and testing. We use a two-layer fully-connected neural network to test each method, as this corresponds to the same network that agents use to encode each others' messages as part of their observations.

Table 3 shows the classification results for the two probing tests. For goal visibility, the easier task, all methods' messages can be effectively used to interpret whether a goal is visible in the observations or not. In the more difficult prediction of goal location, all methods perform above chance (20%) but CACL's protocol significantly outperforms baselines. Contrastive learning across different agents' messages can enable CACL to learn a more global understanding of location. By encoding the goal's spatial information, CACL agents are more likely able to move directly towards it, and reduce episode length. If other methods simply communicate that a goal is found, agents know to change their search but are not as precise in direction. This explains why AEComm, PL, and DIAL perform better than IAC but worse than CACL, which also learns much quicker as shown in Figure 2. For completeness, we also provide classification results with a one-layer (linear) probe with similar results in Appendix A.4

## 5.6 ZERO-SHOT CROSS-PLAY

Table 4: Zero-shot cross-play performance in Predator-Prey. Intra-method results are bolded.

|              | CACL                | AEComm              | PL                  | DIAL                |
|--------------|---------------------|---------------------|---------------------|---------------------|
| CACL (Ours)  | $\mathbf{-17.20 \pm 5.14}$ | $-28.49 \pm 2.78$ | $-24.61 \pm 5.77$ | $-28.78 \pm 3.99$ |
| AEComm       |                     | $\mathbf{-37.86 \pm 7.20}$ | $-31.56 \pm 3.76$ | $-29.73 \pm 3.66$ |
| PL           |                     |                     | $\mathbf{-27.07 \pm 2.94}$ | $-22.89 \pm 3.98$ |
| DIAL         |                     |                     |                     | $\mathbf{-22.85 \pm 2.04}$ |

Table 5: Zero-shot cross-play performance in Find-Goal. Intra-method results are bolded.

|              | CACL                | AEComm              | PL                  | DIAL                |
|--------------|---------------------|---------------------|---------------------|---------------------|
| CACL (Ours)  | $\mathbf{468.75 \pm 15.32}$ | $471.66 \pm 13.54$ | $487.56 \pm 8.61$ | $488.28 \pm 16.60$ |
| AEComm       |                     | $\mathbf{479.96 \pm 14.96}$ | $440.18 \pm 23.04$ | $472.85 \pm 16.77$ |
| PL           |                     |                     | $\mathbf{492.08 \pm 5.67}$ | $486.41 \pm 10.46$ |
| DIAL         |                     |                     |                     | $\mathbf{476.07 \pm 15.89}$ |

An advanced form of coordination is working with partners you have not seen during training (Hu et al., 2020). Previous work has focused on coordination through actions (Carroll et al., 2019; Lupu et al., 2021) but to our knowledge, no previous work has succeeded in learning a linguistic communication protocol that is robust to zero-shot partners. To assess this advanced robustness, we take trained agents from different methods and random seeds and evaluate them with each other (i.e., zero-shot cross-play) in Predator-Prey and Find-Goal. Given two communication learning methods, $m_1$ and $m_2$, we sample two agents from each method for Predator-Prey and for Find-Goal, we average over sampling two agents from one method and one agent from the other and vice-versa. For intra-method cross-play, $m_1 = m_2$, we evaluate agents that were trained with the same method but from different random seeds, so they have not been trained with each other. For inter-method cross-play, $m_1 \neq m_2$, we sample agents from two different methods and pair them with each other. Each pairing is evaluated for 10 random seeds each with 10 evaluation episodes. Given that agents are trained in self-play (Tesauro, 1994) without regard for cross-play, we expect severe performance dips.

We show mean and standard deviation across random seeds for Predator-Prey and Find-Goal in Tables 4 and 5, respectively. As expected, all pairings take a significant dip in performance when

compared with the main results. Inter-method cross-play performance is particularly bad across all algorithms. However, notably, CACL outperforms other methods in intra-method cross-play, indicating that the protocols learned by CACL are generally more robust even across random seeds. In general, zero-shot linguistic communication is incredibly difficult and our results are quite weak. Still, CACL shows promise and demonstrates that contrastive SSL methods can lead to better zero-shot communication and coordination.

## 5.7 PROTOCOL REPRESENTATION LEARNING WITH REINFORCEMENT LEARNING

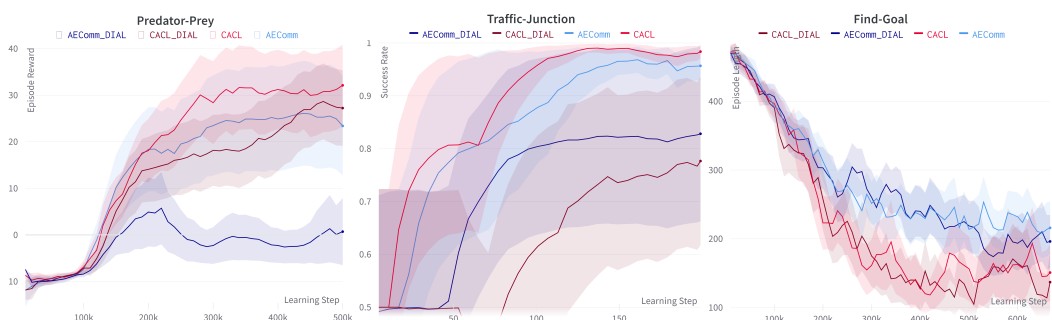

Figure 4: Comparing CACL and AEComm with their respective variants when combined with DIAL. Variants with DIAL have generally worse performance.

Given the overall improved performance of our method, a natural question is whether we can achieve even better results if we use the reward to optimize our message as well. To answer this question, we add DIAL to both CACL and the next best method, AEComm, and evaluate in the three environments. This is equivalent to backpropogating $L_{RL}$ from Equation 2 through agents to learn the message head. In this way, both RL and SSL (contrastive or autoencoding) signals are used to learn the protocol.

Figure 4 compares the performance of CACL and AEComm with their DIAL-augmented variants. We observe that augmenting either SSL method with DIAL performs generally worse, except in Find-Goal, where performances is similar but not better. These findings are consistent with Lin et al. (2021), who find that mixing SSL and RL objectives are detrimental to performance. We hypothesize that decentralized DIAL is a complex, and high-variance optimization that is difficult to stabilize. DIAL's gradient updates may clash with CACL and result in neither a useful contrastive representation, nor a strong reward-oriented one.

## 6 CONCLUSION AND FUTURE WORK

In this work, we introduce an alternative perspective in learning to communicate in decentralized MARL by considering the relationship between messages sent and received within a trajectory. Inspired by multi-view learning, we propose to ground communication using contrastive learning by considering agents' messages to be encoded views of the same state. First, we empirically show that our method leads to better performance and a more consistent, common language among agents. Then, we qualitatively and quantitatively probe our messages as learned representations to show that our method more consistently captures task-relevant information. We also test our method on zero-shot cross-play, a first for MARL communication, and demonstrate promising results. Finally, we show that our SSL objective is not improved by further optimizing with RL, in line with previous work. We believe this work solidifies SSL as an effective method for learning to communicate in decentralized MARL. Furthermore, we have demonstrated a link between multi-view SSL which has been focused on images and communicative MARL. We hope this inspires more investigation at the intersection of these two research directions.

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

# A  APPENDIX

## A.1  ENVIRONMENT DETAILS

Figure A.1 provides a visual illustration of the environments used.

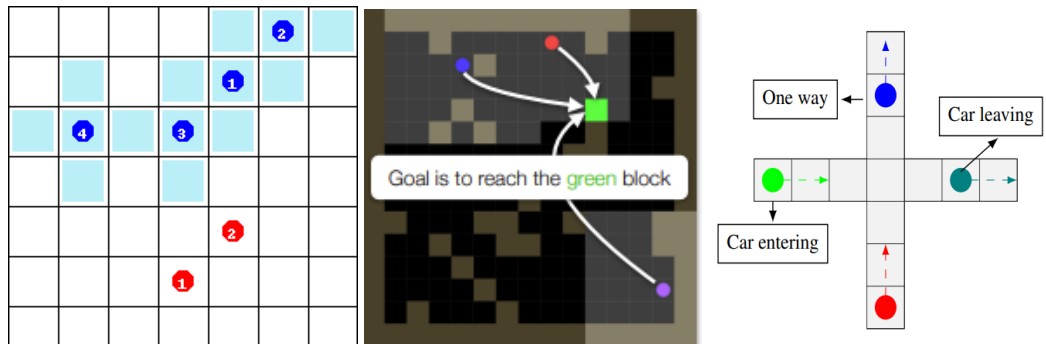

Figure 5: Visual illustration of the environments used. Left: Predator-Prey, taken from Koul (2019). Middle: Find-Goal, taken from Lin et al. (2021). Right: Traffic-Junction, taken from Singh et al. (2018)

### A.1.1  PREDATOR-PREY

We modify the Predator-Prey implementation by Koul (2019). Our Predator-Prey has a higher communication and coordination requirement than the original Predator-Prey environment. Specifically, for a prey to be captured, it has to be entirely surrounded (i.e. the prey cannot move to another grid position in any actions).

Here, we use an 7x7 gridworld. In each agent's observation, it can only see the prey if it is within the field of view (3x3) and cannot see where other agents are. A shared reward of 10 is given for a successful capture and A penalty of -0.5 is given for a failed attempt. A -0.01 step penalty is also applied per step. Each agent has the actions of *LEFT*, *RIGHT*, *UP*. *DOWN* and *NO-OP*. The prey has the movement probability vector of $[0.175, 0.175, 0.175, 0.175, 0.3]$ with each value corresponding to the probability of each action taken.

All algorithms are trained for 30 million environment steps with a maximum of 200 steps per episode.

### A.1.2  FIND-GOAL

We use the Find-Goal environment implementation provided by Lin et al. (2021). The agents have the goal to find where the goal is in a 15x15 grid world with obstacles.

Unlike in Lin et al. (2021), each agent has a 3x3 field of view (instead of 7x7) to make the task more difficult. Each agent receives a reward of 1 for reaching a goal and an additional reward of 5 if all agents reach the goal. We use a step penalty of -0.01 and an obstackle density of 0.15.

All algorithms are trained for 40 million environment steps with a maximum of 512 steps per episode.

### A.1.3  TRAFFIC-JUNCTION

We use the Traffic-Junction environment implementation provided by Singh et al. (2018). The grid-world is 8x8 with 1 traffic junction. The rate of cars being added has a minimum and maximum of 0.1 and 0.3. We use the easy version with two arrival points and 5 agents. Agents are heavily penalized if a collision happens and have only two actions, namely *gas* and *brake*.

All algorithms are trained for 20 million environment steps with a maximum of 20 steps per episode.

## A.2 ARCHITECTURE AND HYPERPARAMETERS

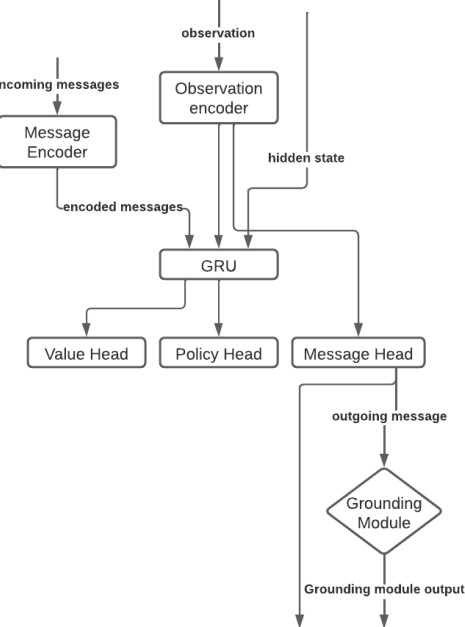

Figure 6: Architectural illustration for algorithms with communication. To remove communication, the message head is disabled. Grounding module is only relevant to CACL and AEComm. The former is a loss function and the latter is a decoder to reconstruct the encoded observation.

Figure 6 illustrates the components of the architecture used in this work, similar to (Lin et al., 2021). A message head is only used for algorithms with communication, namely CACL, AEComm, PL and DIAL. The Grounding Module refers to mechanisms to ground the messages produced by the message head, used in CACL and AEComm. Unless specified otherwise, we fix all hidden layers to be a size of 32.

We experimented with using the output of the GRU, or hidden state, to condition the message head. Empirically we found that directly conditioning on the observation encoding, as in Lin et al. (2021), lead to more stable learning dynamics.

The observation encoder output values of size 32. For Predator-Prey and Traffic Junction, a one-layer fully-connected neural network is used as observation encoder. For Find-Goal, same as Lin et al. (2021), we use a two-layer convolutional neural network followed by a 3-layer fully-connected neural network.

For the message encoder, it outputs values of size 8 in Predator-Prey and Find-Goal with one hidden layer. It outputs values of size 16 in Traffic-Junction with two hidden layers. These configurations are selected based on the best performance of the baseline communication learning algorithm used - DIAL. Messages received are concatenated before passing to message encoders. For all the methods with communication, they produce messages of length 4 ($D = 4$) with a sigmoid function as activation. All models are trained with the Adam optimizer (Kingma & Ba, 2014).

Table 6 lists out the hyperparameters used for all the methods.

## A.3 POSITIVE LISTENING

This section describes the loss function we implemented for positive listening, based on Eccles et al. (2019). Given two policies $\pi^i$ and $\overline{\pi^i}$ of agent $i$ where the latter is the policy with messages zeroed out in the observations, and a trajectory $\tau$ of length $T$, the positive listening loss is written as:

| | |
|---|---|
| Learning Rate | 0.0003 |
| Epsilon for Adam Optimizer | 0.001 |
| $\gamma$ | 0.99 |
| Entropy Coefficient | 0.01 |
| Value Loss Coefficient | 0.5 |
| Gradient Clipping | 2500 |
| $\eta$ for *CACL* | 0.1 |
| $\kappa$ for *CACL* | 0.5 |
| Loss Coefficient for PL | 0.01 |
| Number of Asynchronous Processes | 12 |
| N-step Returns | 5 |

Table 6: Table for hyperparameters used across methods

$$L_{PL} = -\frac{1}{|T|} \sum_{j}^{T} \left[ \sum_{a \in A^i} (|\pi^i(a|\tau_j^i) - \overline{\pi^i}(a|\tau_j^i)|) + (\pi^i(a|\tau_j^i) \log(\overline{\pi^i}(a|\tau_j^i))) \right] \quad (4)$$

where in inner summation, the first term is the L1 Norm and the second term is the cross entropy loss.

### A.4 PROTOCOL REPRESENTATION PROBING: 1-LAYER

Table 7: Classification results of the two probing tests in the Find-Goal environment, comparing all methods with communication. 1-layer neural networks are used for probing

| | DIAL | PL | AEComm | CACL |
|---|---|---|---|---|
| Goal Visibility Test | $94.21\% \pm 2.68$ | $96.93\% \pm 3.14$ | $96.27\% \pm 3.98$ | $87.65\% \pm 3.86$ |
| Goal Location Test | $52.29\% \pm 5.25$ | $53.65\% \pm 9.60$ | $48.16\% \pm 7.34$ | $79.18\% \pm 5.63$ |

Table 7 shows the same results for the two probing tests in section 5.5 except here we use a 1-layer neural network instead of 2 layers. We observe significant dips in performance across all methods. Particularly, CACL becomes worse than the baselines in the easier Goal Visibility Test. However, CACL remains superior in the more difficult Goal Location test by an even bigger margin than the results in table 3.

