# OpenReview forum: "Learning to Communicate using Contrastive Learning "
_ICLR.cc/2023/Conference — Submitted to ICLR 2023_

### Official Review · Reviewer_Tpss · 2022-10-23

**Confidence:** 4
**Correctness:** 2
**Technical Novelty And Significance:** 2
**Empirical Novelty And Significance:** 2
**Recommendation:** 3

**Clarity, Quality, Novelty And Reproducibility:**

the use of the terms linguistics/language/communication is not clear.
there are numerous works on the combination between contrastive loss and RL loss.
the comparison to previous works is implicit evidence only - because the works from 5-6 ago
utilized simpler / smaller architectures and  less computational resources.



**Strength And Weaknesses:**

Strength:
The problem of linguistic communication between decentralized agents is important

Weaknesses:
1.  "Linguistics studies language and its structure, including morphology, syntax, phonetics, and semantics". I do not see any of these in the paper, even though "linguistic communication", linguistics and similar terms are used to describe the advantages of this paper and novelty. There is not language and not linguistics, but the standard multiply view constrictive learning of observations.

2. contrastive (and multi-view) losses can be useful in many scenarios. this paper shows one more such scenario. I am not sure this is enough for a full paper in ICLR. In my opinion the contribution to the knowledge and understanding of linguistic communication for decentralized RL is minor.

3. The current comparison between different approaches is indirect evidence only, because a) there can be different number of parameters in DNNs, (which is not mentioned), b) with more powerful harsher one can achieve better results (more training..).

4. there are no conclusions but rather a summary of the paper.



a few minor comments regarding the phrasing:

1. "..but many works focus on simple, non-MDP environments." what simple non-MDP environments do you mean?
2. "...requires fewer assumptions about other agents..." which assumptions about other are assumed in centralized and decentralized setting? that will help to understand "..fewer..". It is desired to provide evidence.

**Summary Of The Paper:**

this paper proposes to develop linguistic communication between decentralized agents via multiple view contractive learning. which is an interesting problem. it is suggested to combine the contractive loss with the RL loss with a predefined hyperparameter.

**Summary Of The Review:**

there is not linguistics/linguistic-based communication. the objective is standard. there is no general  principles/conclusions.

---

> ### Author Response · Authors · 2022-11-09
> **Clarifications for Reviewer Tpss**
>
> ## Clarifications
>
> **"..but many works focus on simple, non-MDP environments." what simple non-MDP environments do you mean?**
>
> We are referring to environments like Lewis Games in which agents are not situated and play simple referential games (e.g. auto-encoding a symbolic input). These are commonly used in assessing communication in computational agents (i.e., MARL communication). We have rephrased and added references to such works in the updated paper.
>
> **"...requires fewer assumptions about other agents..." which assumptions about other are assumed in centralized and decentralized setting? that will help to understand "..fewer..". It is desired to provide evidence**
>
> Please refer to [1] for more in-depth discussions of the limitations of centralized training. For centralized execution, it requires that all agents be able to communicate in a timely fashion with a centralized controller. This may be infeasible if the agents are very far apart, have unstable connections, or need to send large amounts of information to the central controller on a small bandwidth. It also assumes there are not too many agents, otherwise the central controller may get overwhelmed.
>
> We have clarified our point in the paper and added references to two decentralized MARL environments based on the real world to our paper (controlling ACs [2] and UAVs [3])
>
> [1] Wenhao Li, Bo Jin, Xiangfeng Wang, Junchi Yan, and Hongyuan Zha. F2a2: Flexible fully decentralized approximate actor-critic for cooperative multi-agent reinforcement learning. arXiv preprint arXiv:2004.11145, 2020.
>
> [2] Vincent Mai, Tianyu Zhang, and Antoine Lesage-Landry. Multi-agent reinforcement learning for
> renewable integration in the electric power grid. In Tackling Climate Change with Machine Learn-
> ing Workshop at NeurIPS, 2021.
>
> [3] Soyi Jung, Won Joon Yun, Joongheon Kim, and Jae-Hyun Kim. Coordinated Multi-Agent Deep
> Reinforcement Learning for Energy-Aware UAV-Based Big-Data Platforms. Electronics, 10(5):
> 543, January 2021. ISSN 2079-9292. doi: 10.3390/electronics10050543. URL https://
> www.mdpi.com/2079-9292/10/5/543. Number: 5 Publisher: Multidisciplinary Digital
> Publishing Institute.
>
>
> **the works from 5-6 ago utilized simpler / smaller architectures and less computational resources.**
>
> Our work uses relatively humble resources compared to many modern environments. A single run on a low-end GPU or 8 high-end CPUs takes approximately 5-6 hours. We struggle to find other work on similar MARL environments that is simpler.
>
> **it is suggested to combine the contractive loss with the RL loss with a predefined hyperparameter.**
>
> We would like to emphasize the subtle point that we are not “combining” the contrastive loss and RL loss in Equation 2. These losses are used to train separate networks (communication head, and encoder + policy head respectively) and so they do not actually interact. More importantly, in section 5.7, we showed that the combination of RL and contrastive losses for learning the communication head would worsen performances, consistent with results from previous works.

---

> ### Author Response · Authors · 2022-11-09
> **Response to Reviewer Tpss**
>
> We appreciate the reviewer’s time spent reviewing our work and are glad that they find the problem “interesting” and “important”. However, respectfully, we think the reviewer has misunderstood the general direction and setting of the paper. This work is in the field of communication in (decentralized) multi-agent reinforcement learning, not linguistics. We respond to criticism below and request the reviewer to better detail the issues they have with content on MARL communication and what they believe is insufficient.
>
> ## Major Issues
>
> **"Linguistics studies language and its structure, including morphology, syntax, phonetics, and semantics". I do not see any of these in the paper, even though "linguistic communication", linguistics and similar terms are used to describe the advantages of this paper and novelty. There is not language and not linguistics, but the standard multiply view constrictive learning of observations.**
>
> The reviewer has a misunderstanding of the term “linguistic communication” which is widely used in emergent communication literature [1]. It refers to learned agent communication over a cheap-talk channel. We define this term in the introduction:
>
> > In multi-agent reinforcement (MARL) (Sutton & Barto, 2018), agents can use their actions to transmit information (Grupen et al., 2020) but linguistic communication, in the form of continuous or discrete messages (Foerster et al., 2016), is more flexible and powerful.
>
> Linguistic communication does not refer to natural language and linguistics is outside the scope of this work. We have rephrased this to be even more clear in the updated paper and we can change all references of “language” to “protocol” if that would make things clearer.
>
> [1] Lazaridou, Angeliki, and Marco Baroni. “Emergent Multi-Agent Communication in the Deep Learning Era.” arXiv, July 14, 2020.
>
> **2. contrastive (and multi-view) losses can be useful in many scenarios. this paper shows one more such scenario. I am not sure this is enough for a full paper in ICLR. In my opinion the contribution to the knowledge and understanding of linguistic communication for decentralized RL is minor.**
>
> **… there are numerous works on the combination between contrastive loss and RL loss.**
>
> We strongly disagree. Just because contrastive learning has been used successfully in other, unrelated fields, does not diminish the novelty of our work. Furthermore, to our knowledge there are no other works that use contrastive learning in MARL for representation learning, much less learning to communicate. Next, the perspective of agent’s egocentric views as contrastive views is also novel to our understanding. Finally, our method achieves better performance than all previous baselines (two of which are published papers at NeurIPS on decentralized communication) which seems more than sufficient for ICLR.
>
> The reviewer claims that there are numerous other works that combine contrastive losses with RL in a similar fashion. We would be happy to cite and explain how our work is novel if the reviewer were to point us towards this related work.
>
> **3. The current comparison between different approaches is indirect evidence only, because a) there can be different number of parameters in DNNs, (which is not mentioned), b) with more powerful harsher one can achieve better results (more training..).**
>
> We do not understand what the specific issue here is. This work follows best practices in MARL evaluation to fairly and soundly compare our method with our baselines. All compared methods use the same network architecture and the same number of parameters, accompanied by extensive hyperparameter tuning of baselines. Our architecture is extensively detailed in the appendix. If the reviewer has specific and valid concerns regarding our results, we are happy to further clarify and address them.
>
> **4. there are no conclusions but rather a summary of the paper.**
>
> We have summarized our contributions and due to space constraints detailed one key takeaway for future directions i.e. the further exploration of contrastive learning / SSL methods in MARL. Can the reviewer elaborate on what they believe is missing?

---

> ### Author Response · Authors · 2022-11-15
> **Follow-up**
>
> As we near the end of the discussion period, we would be grateful if the reviewer can let us know their thoughts on our response and changes to the paper. We are keen to address any concerns relevant to the content of our submission before the discussion period ends.

---

### Official Review · Reviewer_dj4f · 2022-10-23

**Confidence:** 5
**Correctness:** 3
**Technical Novelty And Significance:** 3
**Empirical Novelty And Significance:** 3
**Recommendation:** 3

**Clarity, Quality, Novelty And Reproducibility:**

There is still a lot of room for improvement in the clarity of the paper.

It is an exciting job to make agents communicate using contrastive learning in a decentralized setting. While compared with the workshop paper “LEARNING TO GROUND DECENTRALIZED MULTIAGENT COMMUNICATION WITH CONTRASTIVE LEARNING”,  the current version is updated less in the method.

As for Reproducibility,  I believe that it is easy to reproduce the results of the paper reported for the reasons that all technologies adopted in this paper are all used in other related works.

**Strength And Weaknesses:**

 The paper proposes a communication method based on supervised contractive learning with sufficient experiments to evaluate the proposed method.

The weakness of this paper:
1. The problem investigation in Section III is confusing, especially in the usage of symbols with lower and upper case, such as s and S for real states.
2. Meanwhile, is it right for the transition function P->SxAxS?
3. Some citations are in the wrong formation, such as the caption of Fig. 3 for DBSCAN and t-SNE.
4. Compared with the workshop paper “LEARNING TO GROUND DECENTRALIZED MULTIAGENT COMMUNICATION WITH CONTRASTIVE LEARNING”,  the current version is updated less in the method.

**Summary Of The Paper:**

In the paper, the authors propose to communicate between different agents using contrastive learning in a decentralized setting.

As their main contribution of making agents communicate based on contractive learning for multi-agent reinforcement learning in the decentralized setting, they evaluate the proposed model in different environments and obtain promising results.

**Summary Of The Review:**

It is an exciting job to make agents communicate using contrastive learning in a decentralized setting. However, the current version is not suitable for ICLR.

---

> ### Author Response · Authors · 2022-11-09
> **Response to Reviewer dj4f**
>
> We would like to thank the reviewer for their time and are glad the reviewer finds our research direction “exciting”, believes we have performed “sufficient experiments to evaluate the proposed method”, and finds our work detailed enough that it is “easy to reproduce the results of the paper”. We also plan to release our code upon acceptance.
>
> However, we struggle to understand why the reviewer has recommended rejection. Respectfully, we believe all the mentioned weaknesses are either superficial or invalid and the review content to be below ICLR’s standard. We would like to request that the reviewer provide more concrete, useful feedback related to the content of our submission. We respond to each criticism below:
>
> **1. The problem investigation in Section III is confusing, especially in the usage of symbols with lower and upper case, such as s and S for real states.**
>
> **2. Meanwhile, is it right for the transition function P->SxAxS?**
>
> Our problem formulation is based on widely used RL and MARL formulations, as in seminal MARL work like QMIX [1]. The transition function is the exact standard in RL taken from [2]. Does the reviewer have specific issues or suggestions for what to change or what was unclear in Section 3?
>
> [1] Rashid, Tabish, Mikayel Samvelyan, Christian Schroeder, Gregory Farquhar, Jakob Foerster, and Shimon Whiteson. "Qmix: Monotonic value function factorisation for deep multi-agent reinforcement learning." In International conference on machine learning, pp. 4295-4304. PMLR, 2018.
>
> [2] Sutton, Richard S., and Andrew G. Barto. Reinforcement learning: An introduction. MIT press, 2018.
>
> **3. Some citations are in the wrong formation, such as the caption of Fig. 3 for DBSCAN and t-SNE.**
>
> We have fixed the brackets on these citations, please see the updated paper.
>
> **4. Compared with the workshop paper “LEARNING TO GROUND DECENTRALIZED MULTIAGENT COMMUNICATION WITH CONTRASTIVE LEARNING”, the current version is updated less in the method.**
>
> Is the reviewer implying that the workshop paper detracts from the novelty of this archival conference submissions? The non-archival workshop paper should not be of consideration in assessing this work. In machine learning, presenting non-archival workshop papers of ongoing work is standard and does not affect the novelty of submissions for published venues.
>
> **"Current version is not suitable for ICLR"**
>
> None of the reviewer’s critiques have at all addressed the actual content of our work. The reviewer has not challenged any experiments, results, or conclusions of our work. This work is a novel view of communication in multi-agent RL through the lens of contrastive learning. To our knowledge, there is no related work on contrastive learning in MARL, nor any work on learning to communicate with contrastive learning. In that sense, our method and approach are completely novel and our results are strong. On three solid benchmarks, we show that CACL outperforms all baseline methods, including two methods in this research direction from published papers at NeurIPS.
>
> If the reviewer is recommending rejection, and has full confidence in their decision, would they be able to explain specific issues with the content of the paper? We would be happy to address and improve on any relevant concerns and raise the following questions:
>
> 1. Since the direction is “exciting”, why does the reviewer find that the proposed method does not make sufficient progress on this research direction?
> 2. Could the reviewer specifically point out ideas or experiments in the paper that need further improvement in clarity?

---

> ### Author Response · Authors · 2022-11-15
> **Follow-up**
>
> As we near the end of the discussion period, we would be grateful if the reviewer can let us know their thoughts on our response and changes to the paper. We are keen to address any concerns relevant to the content of our submission before the discussion period ends.

---

### Official Review · Reviewer_xPcm · 2022-10-25

**Confidence:** 4
**Clarity, Quality, Novelty And Reproducibility:** Detailed in the main review.
**Correctness:** 3
**Technical Novelty And Significance:** 2
**Empirical Novelty And Significance:** 1
**Recommendation:** 3

**Strength And Weaknesses:**

### Strengths:

- The authors provide a thorough analysis of their proposed method through various embedding probing techniques:
    - qualitative plots for different scenarios,
    - quantitative measures of similarities for messages,
    - transfer of message encodings to auxiliary tasks
- The authors also do a great job of comparing their method to well-known techniques in the literature and show how their approach is better than the others for a wide range of measures
- The section on zero-shot crossplay and the associated analysis was quite interesting to read!
- **************************Clarity:************************** The paper, the methods, the metrics were quite easy to read and understand.

### Weaknesses:

- I think the paper's main weakness lies in justifying the approach's motivation for the relevant problems. I found the introduction and discussion relatively weak in their description of the broader motivations of the work, particularly those relevant to emergent communication: applications/ transfers to the real world and understanding human communication.
- More importantly, similar to Lin et al. (2021), it wasn't clear if this was actually communication; sending the complete representation of the observation without accounting for the other agent as there is no bottleneck. A potential baseline that could test this could use a contrastive loss to learn observation embeddings (w/o multiagent dynamics), freeze that and train agents to communicate by using these. It would help us understand how the setting of sending messages and communicating changes representations.
- **************************Novelty:************************** The approach essentially combines Lin et al. (2021) and with a contrastive info-NCE objective to learn representations.

### Questions:

- In the crossplay setting (Table 4), the interplay agent outperforms the intraplay agent for CACL-AEComm and CACL-PL. Why do the authors think that the interplay agent outperforms agents specifically trained to communicate with each other?
- Is there a systematic split between the training tasks for the agents and the evaluation ones?

**Summary Of The Paper:**

The paper proposes a self-supervised learning objective using a contrastive loss to learn a medium/language for communication between agents to complete a task (in a decentralized training setting). Similar to Lin et al. (2021), the messages here are proposed to be encodings of observations. To learn these encodings/ messages, the contrastive objective uses positive anchors sampled from the current trajectory around a window and negative samples from other messages in the batch. The authors test their method (CACL) on three MARL tasks: Predator-Prey, Find-Goal, and Traffic Junction. Through different analyses for probing representations and testing for zero-shot coordination, the authors present their method's effectiveness.

**Summary Of The Review:**

The paper is thorough in comparing with baselines and probing to understand the representations learned by the proposed method. The paper lacks a proper discussion/justification for the setting, the impact of the method used, its implications for different tasks or the emergence of language.

---

> ### Author Response · Authors · 2022-11-09
> **Clarifications for Reviewer xPcm**
>
>
> ## clarifications
>
> **Baseline observation encoding without multi-agent dynamics**
>
> Agents are indeed communicating and there is a bottleneck. Each agent’s observation is an agent’s position and its FOV e.g. 3x3x3 in predator-prey, whereas the communication is a vector of dimension 8 and therefore is a lossy encoding / bottleneck. As well, the communication is learned on top of the observation encoding (see Figure 6 in appendix).
>
> Learning an observation encoding without multi-agent dynamics is an interesting idea. We believe AEComm already shows this as it uses auto-encoding which learns only a single agent’s view and isn't affected by agents sending/receiving messages. Contrastive methods in vision generally use image data augmentations instead of other agent views so they could work. But they are effective for natural images and would likely lead to bad representations for our discrete, vectorized environment. If the reviewer has a specific offline contrastive method in mind or we have misunderstood the idea, let us know and we would gladly implement it.
>
>
> **In the crossplay setting (Table 4), the interplay agent outperforms the intraplay agent for CACL-AEComm and CACL-PL. Why do the authors think that the interplay agent outperforms agents specifically trained to communicate with each other?**
>
> This is a nice observation. We think this is due to lower symmetry / consistency in baseline methods as compared to CACL. Baseline methods do not regularize their communication and so agents learn very unique protocols between themselves that are not as resilient to swaps, much less inter-play. As well, we believe that CACL’s implicit regularization towards a consistent protocol also creates agents that understand a slightly larger distribution of protocols (as they would understand slight deviations from the consistent protocol).
>
> **Is there a systematic split between the training tasks for the agents and the evaluation ones?**
>
> We follow standard practices in MARL evaluation, where there is no notion of tasks (each environment is learned and tested separately from others) and systematic generalization is not usually tested. Still, we are confident in our agents ability to generalize because we test sufficient random seeds that affect agents’ initial positions, targets’ positions, etc.. which covers the overall distribution of the task.

---

> ### Author Response · Authors · 2022-11-09
> **Response to Reviewer xPcm**
>
> We would like to thank the reviewer for their comments and critiques. We are glad they believe we did “a thorough analysis of [the] proposed method” in “comparing [our] method to well-known techniques in the literature” and “show how [our] approach is better than the others”. Furthermore, we’re glad the reviewer found our zero-shot crossplay experiments “interesting to read” and overall found our paper very clear, “quite easy to read and understand.”
>
> ## major issues
>
> The main issue seems to be **motivation/discussion/justification for the setting and method**, as well as **the impact of the method used, its implications for different tasks or the emergence of language**
>
> We are rephrasing our overall motivation and would greatly appreciate specific feedback as to which part of our story seems insufficient.
>
> Emergent communication, as a field, has generally two goals [1]: 1. analytical work on understanding protocols and 2. Improving AI systems. Within the latter, there are works on 2a. learning more human protocols for human-AI collaboration and 2b. Improving AI systems through learning a protocol for machine-machine communication. We focus on the latter and our overall goal is to learn better machine protocols (e.g. TCP/IP, IoT) to coordinate AI without regard to human communication.
>
> We focus on decentralized learning because it is more flexible and scalable, and therefore more applicable to many real-world scenarios [2]. We view communication as a useful encoding of an agent’s observations. Previous methods did nothing to regulate different agent protocols and lead to each agent learning their own protocol (see Table 2). Using the perspective of contrastive learning, we can implicitly regulate communication in a decentralized way to learn a more unified protocol and in doing so, make communication more effective allowing agents to perform better on collaborative tasks (see Figure 2).
>
> Our work implies that contrastive learning can be an effective paradigm and perspective for protocol learning, especially in a completely decentralized setting. We believe that future machine-machine interactions (e.g. warehouse robots) can use contrastive-learned protocols to more efficiently accomplish their goals. Furthermore, our work may have implications for the emergence of human language in that contrastive learning seems to be an unexplored model for human language evolution but that is outside the scope of our work.
>
> [1] Lazaridou, Angeliki, and Marco Baroni. “Emergent Multi-Agent Communication in the Deep Learning Era.”
> [2] Wenhao Li, Bo Jin, Xiangfeng Wang, Junchi Yan, and Hongyuan Zha. F2a2: Flexible fully decentralized approximate actor-critic for cooperative multi-agent reinforcement learning.
>
> **CACL's novelty: combining AEComm and contrastive learning**
>
> We would like to strongly argue that CACL is not just a novel method but also a novel perspective. First of all, despite the wealth of papers on multi-view contrastive learning, nearly all of them are in computer vision and have focused on image-based data augmentations as a way to create “views”. The perspective that different agents’ egocentric views of an environment can be used as the “views” for contrastive learning is a novel contribution that connects these two, previously-unrelated fields. To our knowledge, there has been no previous work in MARL that uses contrastive learning for representation learning, let alone work in emergent communication that learns to communicate using contrastive learning.
>
> Even without the novelty of agent view as multi-view, CACL is not a straightforward implementation of AEComm and contrastive learning. The choice of SupCon is a non-trivial but essential choice: we use other agents’ views as positive augmentations but our own agent’s views from the timestep window as positive samples. Next, using a window of timesteps as the heuristic for positive views is another unique design choice. It intelligently exploits the inherent nature of POMDPs by assuming environment states do not change too rapidly and is relatively resilient to choice of window size. Overall, we propose a novel perspective on multi-agent communication and intelligently and effectively implement that vision.

---

> ### Author Response · Authors · 2022-11-15
> **Has our response addressed your concerns?**
>
> As we are near the end of the discussion period, we would be grateful if the reviewer can confirm whether our response has addressed their concerns, and let us know if any issues remain so we can try to address them before the discussion period ends.

---

### Author Response · Authors · 2022-11-09
**Initial paper update**

We have updated the paper with respect to reviewer's initial comments and put all our changes in red.

---

### Decision · Program_Chairs · 2023-01-20

**Decision:**

Reject

**Justification For Why Not Higher Score:**

N/A

**Justification For Why Not Lower Score:**

N/A

**Metareview: Summary, Strengths And Weaknesses:**

This paper proposes using contrastive learning as a mechanism to encourage agents to effectively communicate with each other in a multi-agent setting. This is achieved by encoding an agents observations into the communication channel, but also maximizing the mutual information between messages within an episode. This simple yet effective idea leads to more effective and symmetric communication as has bene demonstrated quantitatively. The authors also demonstrate a slightly surprising result, i.e. zero-shot communication between separately trained agents.

Three authors reviewed this paper and provided reject ratings. After reading the paper myself, reading the reviews and author rebuttal carefully. I disagree with the reviewers in their recommendation to reject the paper.

Reviewer xPcm has two strong concerns:
(1) Positioning of the work (but has failed to point out specific arguments or lines in the paper that they disagree with) --
In response, the authors have provided a detailed rebuttal regarding the positioning of this work, which I believe addresses this concern.

(2) Novelty of the work --
I agree with the authors take on this. The proposed CACL method isnt just novel by itself, but importantly it connects two different sets of problems that people have been working on -- multi agent communication and contrastive learning for representation learning. I think this connection (which now feels natural after reading this paper) is quite interesting and opens up this area for further research. This connection is novel in my opinion, is simple and has been shown to be effective.

Reviewer dj4f has provided one strong concern that I discuss below. Other concerns are minor formatting concerns and in my opinion, should not be grounds for such a strong negative rating. They have also been addressed in the rebuttal.
(1) Previous workshop paper.
The ICLR guidelines state that: "Submissions that are identical (or substantially similar) to versions that have been previously published, or accepted for publication, or that have been submitted in parallel to this or other conferences, journals or archival workshops, are not allowed and violate our dual submission policy."
The workshop that the reviewer is referring to is non-archival. Hence as per ICLR policy, I am disregarding this workshop submission and making my recommendation purely on the merits of the submitted draft.
I consider this concern resolved.

Reviewer Tpss has a few concerns such as a lack of a conclusion (which is certainly not grounds for rejection). The two strong concerns in my opinion are:

(1) Contrastive learning + X is now well studied --
As the authors have stated and I agree, this does not devalue the contributions of this paper. In particular, the use of contrastive learning to improve multi-agent communication is new and effective and an important contribution to this field.
(2) They are opposed to the use of the term "linguistic communication" --
The authors have noted the context of this use, and have updated the paper to make this clear. This is

In summary, the strong reject ratings by the three reviewers do not seem justified. Several concerns are minor ones that have been easily fixed in the revised draft and many have been addressed in the rebuttal. These more superficial concerns are not grounds for rejecting the paper. One main concern is about the novelty of the paper, which I have addressed above and where I agree with the authors. And one more strong concern is about an overlapping submission at a workshop which I also consider resolved.

However, we are currently also missing sufficient agreement amongst domain experts that the paper is ready for acceptance. After detailed discussion between PCs, SAC, and AC, Program Chairs indicated that available feedback and consensus isn’t sufficient for overruling strong signals from reviewers (all strong reject). As such, the paper can not be considered for acceptance in the program of ICLR 2023.

We strongly encourage the authors to revise (e.g., with respect to clarity, as two reviewers pointed out, as well as insights into the proposed method) and resubmit their manuscript as it provides promising contributions for the community

**Summary Of Ac-Reviewer Meeting:**

N/A